# Epitope Mapping of Anti-Mouse CCR3 Monoclonal Antibodies Using Flow Cytometry

**DOI:** 10.3390/antib11040075

**Published:** 2022-12-02

**Authors:** Nami Tateyama, Teizo Asano, Hiroyuki Suzuki, Guanjie Li, Takeo Yoshikawa, Tomohiro Tanaka, Mika K. Kaneko, Yukinari Kato

**Affiliations:** 1Department of Antibody Drug Development, Tohoku University Graduate School of Medicine, Sendai 980-8575, Japan; 2Department of Molecular Pharmacology, Tohoku University Graduate School of Medicine, Sendai 980-8575, Japan; 3Department of Pharmacology, Tohoku University Graduate School of Medicine, Sendai 980-8575, Japan

**Keywords:** mouse CCR3, monoclonal antibody, epitope mapping, alanine scanning, flow cytometry

## Abstract

The CC chemokine receptor 3 (CCR3) is a receptor for CC chemokines, including CCL5/RANTES, CCL7/MCP-3, and CCL11/eotaxin. CCR3 is expressed on the surface of eosinophils, basophils, a subset of Th2 lymphocytes, mast cells, and airway epithelial cells. CCR3 and its ligands are involved in airway hyperresponsiveness in allergic asthma, ocular allergies, and cancers. Therefore, CCR3 is an attractive target for those therapies. Previously, anti-mouse CCR3 (mCCR3) monoclonal antibodies (mAbs), C_3_Mab-3 (rat IgG_2a_, kappa), and C_3_Mab-4 (rat IgG_2a_, kappa) were developed using the Cell-Based Immunization and Screening (CBIS) method. In this study, the binding epitope of these mAbs was investigated using flow cytometry. A CCR3 extracellular domain-substituted mutant analysis showed that C_3_Mab-3, C_3_Mab-4, and a commercially available mAb (J073E5) recognized the N-terminal region (amino acids 1–38) of mCCR3. Next, alanine scanning was conducted in the N-terminal region. The results revealed that the Ala2, Phe3, Asn4, and Thr5 of mCCR3 are involved in C_3_Mab-3 binding, whereas Ala2, Phe3, and Thr5 are essential to C_3_Mab-4 binding, and Ala2 and Phe3 are crucial to J073E5 binding. These results reveal the involvement of the N-terminus of mCCR3 in the recognition of C_3_Mab-3, C_3_Mab-4, and J073E5.

## 1. Introduction

Chemokines are a family of small cytokines secreted by cells, and they play essential roles in cell migration, inflammation, and immune responses by binding to chemokine receptors [1,2,3,4]. The CC chemokine receptor 3 (CCR3) is a receptor for CC chemokines, including CCL5/RANTES, CCL7/MCP-3, and CCL11/eotaxin [5,6,7]. CCR3 is expressed on the surface of eosinophils, basophils, a subset of Th2 lymphocytes, mast cells, and airway epithelial cells [8,9,10,11,12,13]. CCR3 is a family of G-protein-coupled receptors (GPCRs) that transduce extracellular signals to intracellular signaling molecules [14]. The CCR3 signaling pathway is critical in eosinophil migration [15,16]. It has been reported that CCR3 and its ligands can cause airway hyperresponsiveness in a murine allergic asthma model [17,18,19,20], contributing to ocular allergies [21]. Moreover, elevated eotaxin expression has been observed in colorectal cancer [22], breast cancer [23], and oral squamous cell carcinomas [24]. Therefore, CCR3 and its ligands are therapeutic targets for allergic diseases and cancers [7,25].

GPCR has seven transmembrane domains; four extracellular regions, including an N-terminal region (amino acids [aa] 1–38); and three extracellular loops (ECL1; aa 96–111, ECL2; aa 176–207, and ECL3; aa 269–285). Previously, monoclonal antibodies (mAbs) were developed against GPCRs, including an anti-mouse CCR2 mAb [26], an anti-human CCR2 mAb [27], an anti-mouse CCR3 (mCCR3) mAb [28,29,30], an anti-mouse CCR4 mAb [31], an anti-mouse CCR8 mAb [32,33,34], an anti-human CCR9 mAb [35], and an anti-mouse CXCR6 mAb [36]. The binding epitopes of anti-CCR2, CCR4, CCR9, and CXCR6 mAbs, which were established using the peptide immunization method, were determined using an enzyme-linked immunosorbent assay (ELISA) [37,38,39]. However, anti-mCCR3 mAbs, C_3_Mab-3 (rat IgG_2a_, kappa) and C_3_Mab-4 (rat IgG_2a_, kappa), were established using the Cell-Based Immunization and Screening (CBIS) method [30]. We encountered difficulty in determining the epitopes by ELISA in mAbs developed using the CBIS method [40,41]. Therefore, flow cytometry could have better coverage than ELISA for epitope mapping.

In this study, the epitope mapping of anti-mCCR3 mAbs was conducted using flow cytometry, utilizing the extracellular region substitution and the alanine scanning methods to clarify the features of C_3_Mab-3 and C_3_Mab-4.

## 2. Materials and Methods

### 2.1. Cell Lines

Chinese hamster ovary (CHO)-K1 cells were obtained from the America Type Culture Collection (ATCC, Manassas, VA, USA). The CHO/mCCR3 cells were produced in our previous study [28]. The chimera and the point mutant plasmids were transfected into CHO-K1 cells using the Neon Transfection System (Thermo Fisher Scientific Inc., Waltham, MA, USA). Stable transfectants were selected using a cell sorter (SH800; Sony Biotechnology Inc., Tokyo, Japan). The cells were cultured in Roswell Park Memorial Institute (RPMI) 1640 medium (Nacalai Tesque, Inc., Kyoto, Japan) supplemented with 10% heat-inactivated fetal bovine serum (FBS) (Thermo Fisher Scientific Inc.), 100 units/mL of penicillin, 100 μg/mL streptomycin, and 0.25 μg/mL amphotericin B (Nacalai Tesque, Inc.) at 37 °C in a humidified atmosphere containing 5% CO_2_. The stable transfectants were cultivated in a medium containing 0.5 mg/mL Zeocin (InvivoGen, San Diego, CA, USA).

### 2.2. Plasmid Construction

Synthesized DNA (Eurofins Genomics KK, Tokyo, Japan) encoding mCCR3 (Accession No.: NM_009914.4) [28,29,30] and mouse CCR8 (mCCR8; Accession No.: NM_007720.2) [32,33,34] were subcloned into a pCAG-Ble vector (FUJIFILM Wako Pure Chemical Corporation, Osaka, Japan). Chimeric mutants mCCR8 (mCCR3p1-38), mCCR8 (mCCR3p96-111), mCCR8 (mCCR3p176-207), and mCCR8 (mCCR3p269-285) were produced with a RAP [42,43] and a MAP tag [44,45] at their C-terminus using a HotStar HiFidelity polymerase kit (Qiagen Inc., Hilden, Germany). Alanine (glycine) substitutions in the mCCR3 N-terminal region were conducted using QuikChange Lightning Site-Directed Mutagenesis Kits (Agilent Technologies Inc., Santa Clara, CA, USA). PCR fragments bearing the desired mutations were inserted into the pCAG-Ble vector (FUJIFILM Wako Pure Chemical Corporation) using an In-Fusion HD Cloning Kit (TaKaRa Bio, Inc., Shiga, Japan).

### 2.3. Antibodies

C_3_Mab-4 was established using the CBIS method together with C_3_Mab-3 [30]. C_3_Mab-7 was established using N-terminal peptide immunization as described previously [29]. An anti-mCCR3 mAb (clone J073E5) was purchased from BioLegend (San Diego, CA, USA). A secondary Alexa Fluor 488-conjugated anti-rat IgG was also purchased from Cell Signaling Technology, Inc. (Danvers, MA, USA).

### 2.4. Flow Cytometry

The cells were harvested after a brief exposure to 0.25% trypsin/1 mM ethylenediaminetetraacetic acid (Nacalai Tesque, Inc.). After washing with 0.1% bovine serum albumin in phosphate-buffered saline, the cells were treated with primary mAbs (1 μg/mL) for 30 min at 4 °C and subsequently with Alexa Fluor 488-conjugated anti-rat IgG (1:1000; Cell Signaling Technology, Inc., Danvers, MA, USA). Fluorescence data were obtained using an EC800 Cell Analyzer (Sony Biotechnology Inc.).

## 3. Results

### 3.1. Determination of the Epitope of Anti-mCCR3 mAbs Using Flow Cytometry and Chimeric Proteins

C_3_Mab-3 and C_3_Mab-4 were established using the CBIS method, and they are applicable to flow cytometry [29]. To investigate the binding epitopes of C_3_Mab-3 and C_3_Mab-4, we focused on four extracellular regions of mCCR3, namely, the N-terminal region (aa 1-38), ECL1 (aa 96-111), ECL2 (aa 176-207), and ECL3 (aa 269-285). The four extracellular regions of mCCR3 were substituted into the corresponding regions of mCCR8, which possesses an amino acid structure similar to that of mCCR3. As shown in Figure 1, mCCR8 (mCCR3p1-38), mCCR8 (mCCR3p96-111), mCCR8 (mCCR3p176-207), and mCCR8 (mCCR3p269-285) were generated. The chimeric proteins were transiently expressed on the CHO-K1 cells, and their reactivities to C_3_Mab-3, C_3_Mab-4, and commercially available J073E5 were analyzed using flow cytometry. As shown in Figure 2, C_3_Mab-3, C_3_Mab-4, and J073E5 slightly reacted with mCCR8 (mCCR3p1-38). In contrast, they did not react with mCCR8 (mCCR3p96-111), mCCR8 (mCCR3p176-207), or mCCR8 (mCCR3p269-285). They reacted with mCCR3, which was stably overexpressed in CHO-K1 cells. These results show that the N-terminal region of mCCR3 is recognized by C_3_Mab-3, C_3_Mab-4, and J073E5.

### 3.2. Determination of the C_3_Mab-3 Epitope Using Flow Cytometry and Alanine Scanning

Next, alanine scanning was conducted in the N-terminal region, except for Cys28. Thirty-six alanine substitution mutants of mCCR3 were constructed, and the mutant proteins were transiently expressed on the CHO-K1 cells. The reactivity against C_3_Mab-3, C_3_Mab-4, and J073E5 was assessed using flow cytometry. As shown in Figure 3A, C_3_Mab-3 did not react with four mutants (A2G, F3A, N4A, and T5A). In contrast, C_3_Mab-3 reacted with the other 32 mutants. These results show that four residues (Ala2, Phe3, Asn4, and Thr5) of mCCR3 are important for C_3_Mab-3 binding (Figure 3B). C_3_Mab-4 did not react with three mutants (A2G, F3A, and T5A) but reacted with the others (Figure 4A), indicating that three residues (Ala2, Phe3, and Thr5) of mCCR3 are important for C_3_Mab-4 binding (Figure 4B). J073E5 did not react with two mutants (A2G and F3A) but reacted with the others (Figure 5A), indicating that two residues (Ala2 and Phe3) of mCCR3 are important for J073E5 binding (Figure 5B). The cell surface expression of the mCCR3 mutants on the CHO-K1 cells was confirmed using the anti-mCCR3 mAb, C_3_Mab-7. It has already been confirmed that Phe15 and Glu16 are essential for C_3_Mab-7 binding (manuscript submitted). We could confirm the cell surface expression of four mutants (A2G, F3A, N4A, and T5A) of mCCR3 using C_3_Mab-7 (Figure 6).

## 4. Discussion

We previously established various mAbs against membrane proteins using the CBIS method. Because mAbs sometimes recognize conformational epitopes, they can be applied to flow cytometry but not to Western blotting or ELISA. Two anti-mCCR3 mAbs examined in this study, C_3_Mab-3 and C_3_Mab-4, were established using the CBIS method [30]. We attempted to identify their epitopes using synthetic peptides and ELISA. However, they did not recognize the synthetic peptides, including the mCCR3 N-terminal region (aa 1–19), which contains the epitope determined using flow cytometry (Figure 3 and Figure 4). These results suggest that the residues participate in the formation of conformational epitopes and/or undergo post-translational modification on the cell surface. Furthermore, we could not exclude the possibility of the first Met as their epitopes. In the case of C_3_Mab-3 and C_3_Mab-4 epitopes, Asn4 and Thr5 are involved in recognition. Although Asn and Thr are known to be *N*- and *O*-glycosylated, respectively, there are no reports on the glycosylation of the Asn4 and Thr5 of mCCR3. Further studies are required to analyze the involvement of the post-translational modification of these residues in the recognition by C_3_Mab-3 and C_3_Mab-4.

Other anti-mCCR3 mAbs, C_3_Mab-6, and C_3_Mab-7, were also developed via mCCR3 N-terminal peptide immunization [29]. It was found that they could recognize the synthetic peptide of the mCCR3 N-terminal region (aa 1-19) using ELISA and cell-surface-expressed mCCR3 using flow cytometry. Furthermore, the Phe3, Asn4, Thr5, Asp6, Glu7, Lys9, Thr10, and Glu13 of mCCR3 were determined to be C_3_Mab-6 epitope, whereas Phe15 and Glu16 were determined to be C_3_Mab-7 epitope (manuscript submitted). These results indicate that C_3_Mab-6 and C_3_Mab-7 recognize both the naked N-terminal peptide and cell-surface-expressed mCCR3. Further studies are essential to understand the difference in mCCR3 recognition between C_3_Mab-3/C_3_Mab-4 and C_3_Mab-6/C_3_Mab-7.

It has been reported that a CCR3 ligand, CCL11/eotaxin, binds to the N-terminal region of CCR3 [46,47]. Therefore, our established anti-mCCR3 mAbs could compete with ligand binding to mCCR3 and have neutralizing activity. Shen et al. reported that anti-CCR3 mAb could significantly suppress airway eosinophilia and mucus overproduction in asthmatic mice; therefore, the blockage of the CCR3 axis may be an attractive strategy for asthma therapy [48]. In future studies, we would like to examine the neutralizing activities of these anti-mCCR3 mAbs.

## Figures and Tables

**Figure 1 antibodies-11-00075-f001:**
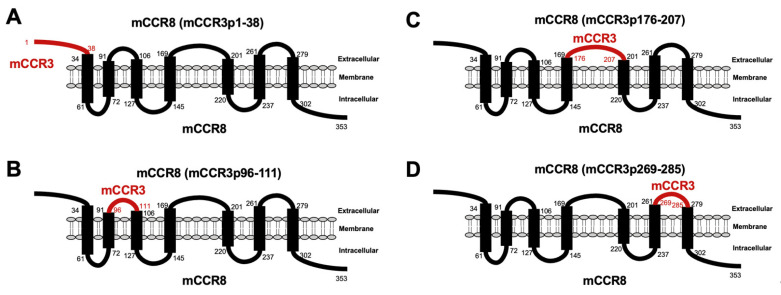
Schematic illustration of mCCR8 and mCCR3 chimeric proteins. The four extracellular regions of mCCR3, namely, (**A**) the N-terminal region (aa 1–38), (**B**) ECL1 (aa 96–111), (**C**) ECL2 (aa 176–207), and (**D**) ECL3 (aa 269–285), were substituted into the corresponding regions of mCCR8. ECL, extracellular loop, aa; amino acids.

**Figure 2 antibodies-11-00075-f002:**
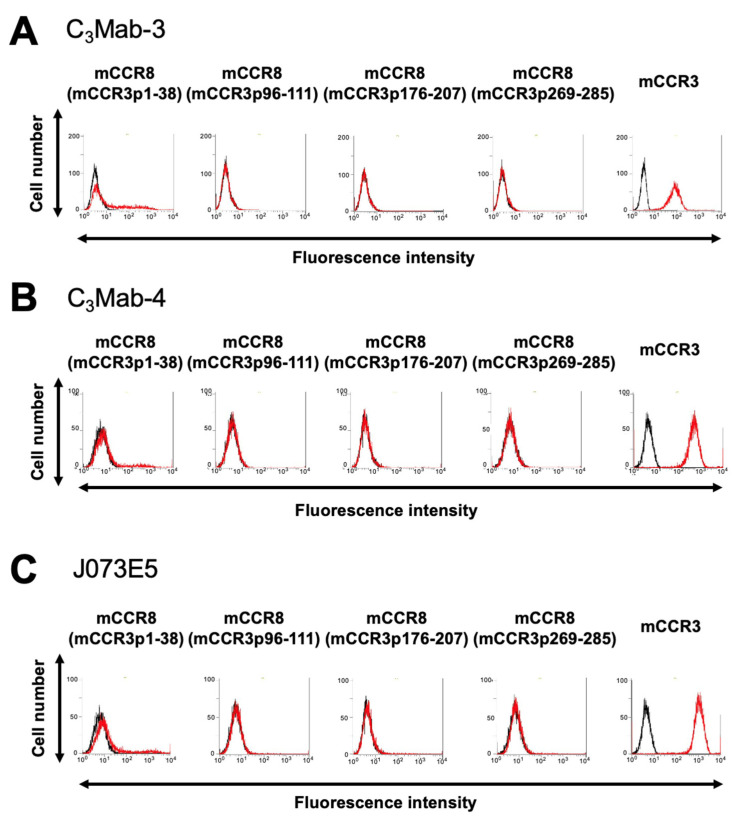
Determination of the epitope of anti-mCCR3 mAbs using flow cytometry and chimeric proteins. C_3_Mab-3 (1 µg/mL) (**A**), C_3_Mab-4 (1 µg/mL) (**B**), and J073E5 (1 µg/mL) (**C**) were treated with CHO-K1 cells that transiently expressed chimeric proteins for 30 min at 4 °C, followed by the addition of Alexa 488-conjugated anti-rat IgG. Red lines show the cells with anti-mCCR3 mAbs treatment, and black lines show cells without anti-mCCR3 mAbs treatment as a negative control.

**Figure 3 antibodies-11-00075-f003:**
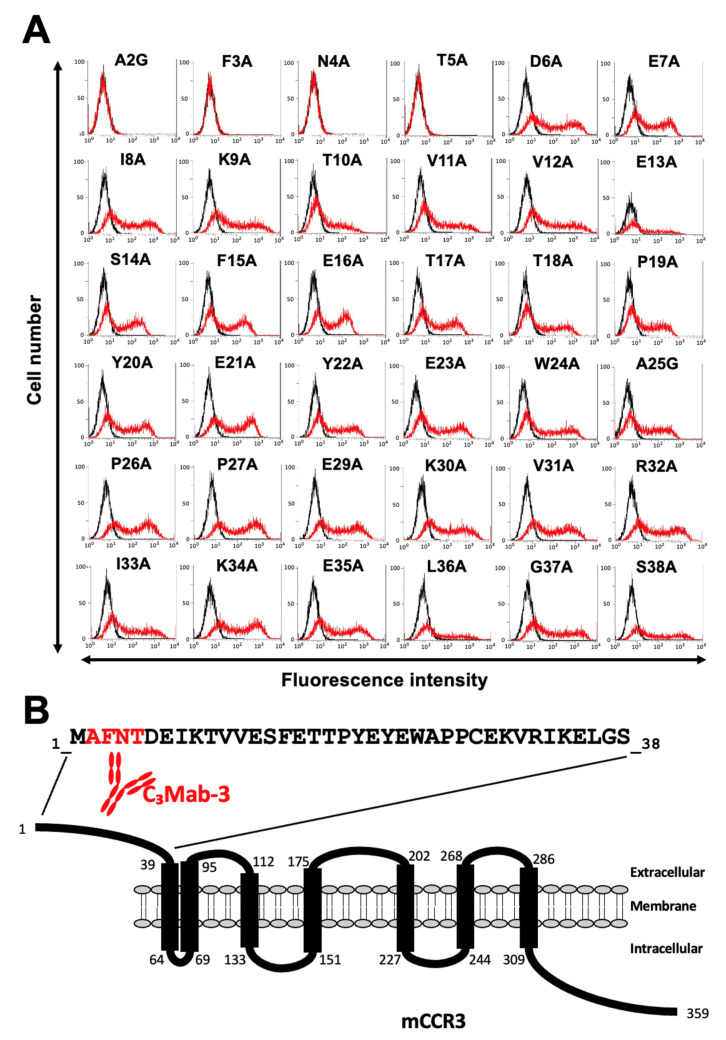
Determination of the C_3_Mab-3 epitope using flow cytometry and alanine scanning. (**A**) C_3_Mab-3 (1 µg/mL) was treated with CHO-K1 cells that transiently expressed mutant proteins for 30 min at 4 °C, followed by the addition of Alexa 488-conjugated anti-rat IgG. Red lines show the cells with C_3_Mab-3 treatment, and black lines show cells without Ab treatment as a negative control. (**B**) The C_3_Mab-3 epitope for mCCR3 involves Ala2, Phe3, Asn4, and Thr5 of mCCR3.

**Figure 4 antibodies-11-00075-f004:**
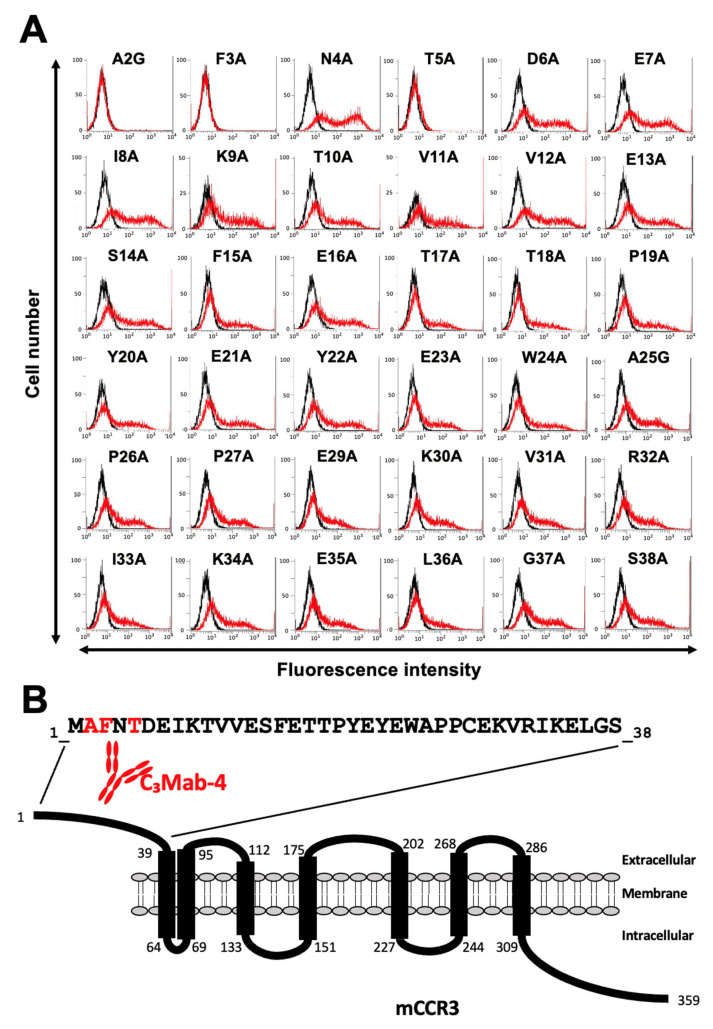
Determination of the C_3_Mab-4 epitope using flow cytometry and alanine scanning. (**A**) C_3_Mab-4 (1 µg/mL) was treated with CHO-K1 cells that transiently expressed mutant proteins for 30 min at 4 °C, followed by the addition of Alexa 488-conjugated anti-rat IgG. Red lines show the cells with C_3_Mab-4 treatment, and black lines show cells without Ab treatment as a negative control. (**B**) The C_3_Mab-4 epitope for mCCR3 involves Ala2, Phe3, and Thr5 of mCCR3.

**Figure 5 antibodies-11-00075-f005:**
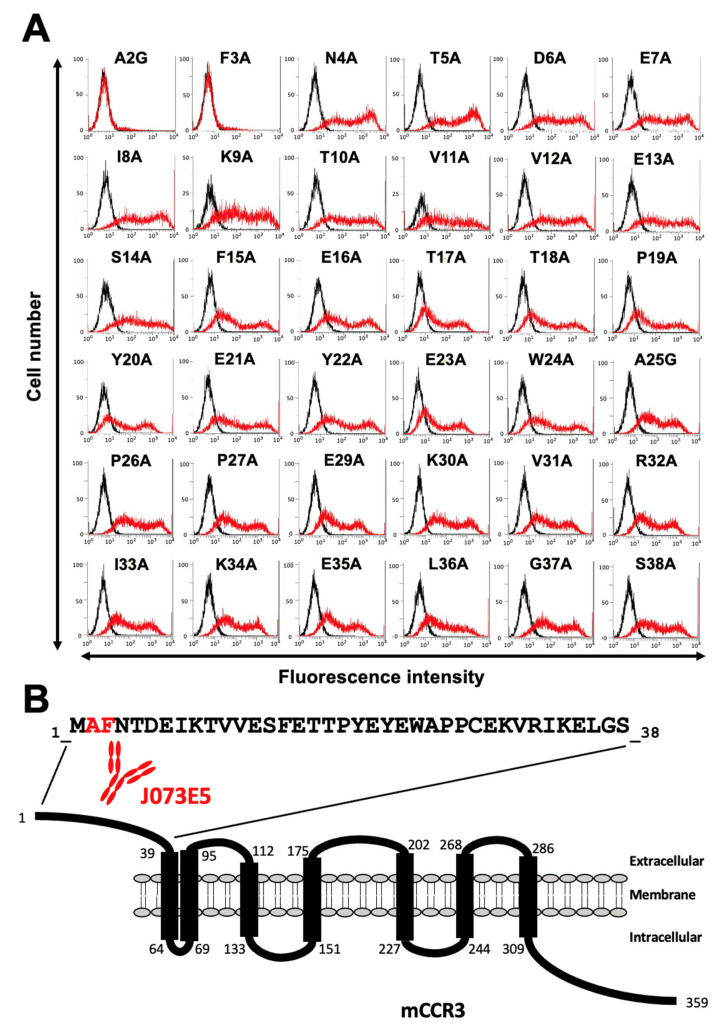
Determination of the J073E5 epitope using flow cytometry and alanine scanning. (**A**) J073E5 (1 µg/mL) was treated with CHO-K1 cells that transiently expressed mutant proteins for 30 min at 4 °C, followed by the addition of Alexa 488-conjugated anti-rat IgG. Red lines show the cells with J073E5 treatment, and black lines show cells without Ab treatment as a negative control. (**B**) The J073E5 epitope for mCCR3 involves Ala2 and Phe3 of mCCR3.

**Figure 6 antibodies-11-00075-f006:**
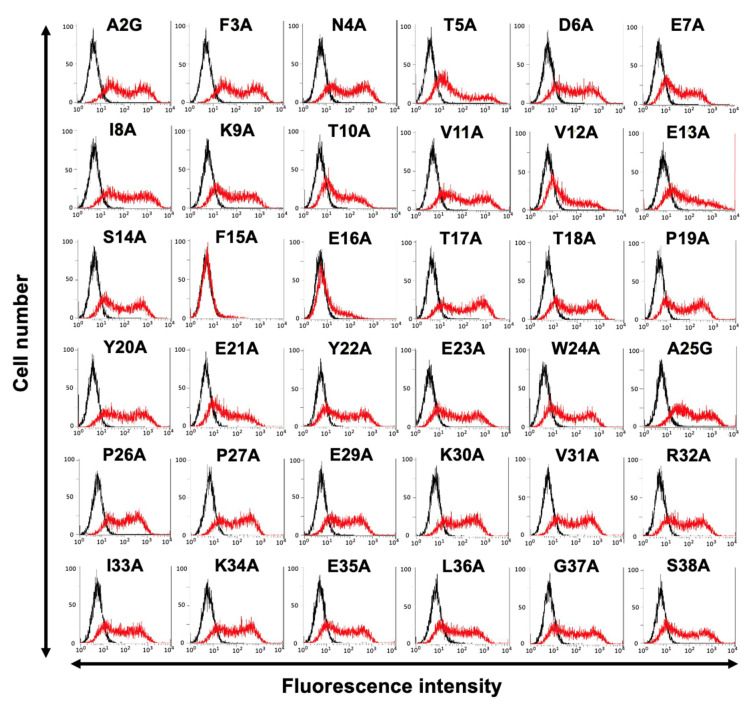
Cell surface expression of mCCR3 mutants on CHO-K1 cells using flow cytometry. C_3_Mab-7 (1 µg/mL) was treated with CHO-K1 cells that transiently expressed mutant proteins for 30 min at 4 °C, followed by the addition of Alexa 488-conjugated anti-rat IgG. Red lines show the cells with C_3_Mab-7 treatment, and black lines show cells without Ab treatment as a negative control.

## Data Availability

Data is contained within the article.

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
