# Peer review of "Epitope Mapping of Anti-Mouse CCR3 Monoclonal Antibodies Using Flow Cytometry"

_2073-4468, 2022, doi:10.3390/antib11040075_

Round 1

Reviewer 1 Report

Brief summary

In this study authors have done the epitope mapping of 2 previously identified monoclonal antibodies for CCR3. They have made multiple mutants of CCR3 and tested binding by flow cytometry to define the epitopes of those antibodies. Though, manuscript is well written and clearly presents the findings, however, justifying following comments will further strengthen the manuscript –

Comments- 

1.       Line 56, Instead of saying “ELISA could not be applied to their epitope mapping” it should be modified to say - Flow cytometry could have better coverage than ELISA for the epitope mapping.

2.       Line 81, why its wrote as Alanine (glycin) substitutions? Maybe typo error?

3.       Line 87, “C3Mab-3 [30] and C3Mab-7 [29] were described previously.”, need to provide brief details for the source of mAbs, in addition to referring with citation.

4.       Line 101, sentence need to be modified. “established using the CBIS method, and are applicable to flow cytometry, but not to ELISA”. Maybe - mAbs are discovered or identified etc by CBIS method, and characterized or etc by flow cytometry. You cannot say “ not applicable to ELISA” without testing it.

5.       Line 110-114, sentence need to be modified, as the binding to mCCR8 (mCCR3p1-38) is not very clear, it looks very week compared to a good binding for mCCR3. This can be discussed a little bit either here or in discussion section.

6.       Line 171. Since, you tested those mAbs by ELISA, it can go in result section.

Author Response

Reviewer1 (authors’ response)

Brief summary

In this study authors have done the epitope mapping of 2 previously identified monoclonal antibodies for CCR3. They have made multiple mutants of CCR3 and tested binding by flow cytometry to define the epitopes of those antibodies. Though, manuscript is well written and clearly presents the findings, however, justifying following comments will further strengthen the manuscript –

Comments- 

  1. Line 56, Instead of saying “ELISA could not be applied to their epitope mapping” it should be modified to say - Flow cytometry could have better coverage than ELISA for the epitope mapping.

We changed as the reviewer suggested.

  1. Line 81, why its wrote as Alanine (glycin) substitutions? Maybe typo error?

Almost amino acids were changed to Alanine. In case of alanine in original sequence, we changed to glycin (e.g. A25G in Fig. 3).

  1. Line 87, “C3Mab-3 [30] and C3Mab-7 [29] were described previously.”, need to provide brief details for the source of mAbs, in addition to referring with citation.

We changed as the reviewer suggested.

  1. Line 101, sentence need to be modified. “established using the CBIS method, and are applicable to flow cytometry, but not to ELISA”. Maybe - mAbs are discovered or identified etc by CBIS method, and characterized or etc by flow cytometry. You cannot say “ not applicable to ELISA” without testing it.

We deleted “ not applicable to ELISA” in the section.

In the discussion, we described that “they did not recognize the synthetic peptides, including the mCCR3 N-terminal region (p1–19), which contains the epitope determined using flow cytometry (Fig. 3 and 4).”

  1. Line 110-114, sentence need to be modified, as the binding to mCCR8 (mCCR3p1-38) is not very clear, it looks very week compared to a good binding for mCCR3. This can be discussed a little bit either here or in discussion section.

We changed.

  1. Line 171. Since, you tested those mAbs by ELISA, it can go in result section.

We have only tested the mCCR3 N-terminal region by ELISA, but not all extracellular regions. Therefore, we added it in the discussion.

Reviewer 2 Report

The citations of the statements on lines 55 and 56 are missing.

 For clarity Figure 2 should include the name of the molecule used (e.g., C3Mab-3, C3Mab-4) above each panel.

In line 138 they mention the interaction of C3Mab-7 with Phe15 and Glu16 residues, which they use to confirm the epitope of C3Mab-7, however, the information is not available for consultation and, therefore it is necessary to present evidence to support the interactions.

Author Response

Reviewer 2 (authors’ response)

The citations of the statements on lines 55 and 56 are missing.

We changed the position of citation.

For clarity Figure 2 should include the name of the molecule used (e.g., C3Mab-3, C3Mab-4) above each panel.

We added.

In line 138 they mention the interaction of C3Mab-7 with Phe15 and Glu16 residues, which they use to confirm the epitope of C3Mab-7, however, the information is not available for consultation and, therefore it is necessary to present evidence to support the interactions.

As shown in Fig.6 (3rd row), the reactivities of C3Mab-7 were almost disapeared in F15A and E16A, indicating that F15 and E16 involve in the recognition by C3Mab-7.

Round 2

Reviewer 2 Report

The authors should present references or some explanation to support that their antibodies obtained by CBIS method are not usable in ELISA.

The authors have satisfactorily solved the other recommendations.

Author Response

We added the description and references, as follows.

Line56

We have faced the difficulty to determine the epitope of mAbs developed by the CBIS method usng ELISA [40,41].